# Assessment of potential land suitability for rainfed wheat production using GIS and multi criteria decision analysis in the Southwestern parts of Ethiopia

**Bacha Gebissa Negeri[1,2], Bai Xiuguang[1]\*, Mitiku Badasa Moisa (ID)[3]**

**1** College of Economics and Management, Northwest A and F University, Yangling, Shaanxi, China,
**2** Department of Agricultural Economics, Faculty of Resource Management and Economics, Wollega University Shambu Campus, Shambu, Ethiopia, **3** Department of Earth Science, College of Natural and Computational Science, Wollega University, Nekemte Campus, Ethiopia

\* baixg960@nwsuaf.edu.cn

## Abstract

Wheat production in Ethiopia is vital for improving food security, boosting the national economy, and achieving self-sufficiency in food consumption. The present study aims to assess the potential land suitability for rainfed wheat (*Triticum aestivum* L.) production by using Geographic Information System and multi criteria decision analysis in southwestern parts of Ethiopia. Biophysical data, including land use and land cover (LULC), soil drainage, soil texture, soil depth, proximity to markets and roads, land surface temperature, slope, rainfall, and elevation, were used. In addition, different software tools, such as ArcGIS 10.3, ERDAS Imagine 2015, IDRISI Selva 17, and ArcSWAT were applied. The results revealed that approximately 177.1 km² (1.3%) of the study area was classified as highly suitable, 5375.2 km² (38.2%) as moderately suitable, 7,246.0 km² (51.5%) as marginally suitable, and 1235.1 km² (8.8%) as currently not suitable for rainfed wheat cultivation. Furthermore, out of the 23 districts analyzed, Sayo Nole and Bedelle Zuriya were identified as highly suitable for wheat production, with an area of 32.7km² and 23.3km² respectively. Therefore, the study recommends that future study research investigate additional other ecological parameters, such as soil PH, lime, gypsum, salinity, alkalinity and socio-economic data, which were not included in the present study.

## 1. Introduction

The report of United Nations Population Division, as cited by [1] predicted that the global population would reach 9.1 billion by 2050. The demand for food in developing countries is rising steadily due to the significant growth in population [2,3]. Population growth, leading to food shortages, often compels people to migrate from rural areas to urban centers in search of better opportunities and access to sustenance [4–6].

**Data availability statement:** All relevant data are within the paper and its Supporting Information files.

**Funding:** The author(s) received no specific funding for this work.

**Competing interests:** The authors declared no conflict of interests

Addressing these challenges requires a strong focus on enhancing agricultural production and productivity. Agricultural production of cereal crops plays a crucial role in global nutrition, serving as a primary source of food calories and protein [7,8]. Ethiopia, one of the least developing countries, produces cereal crops for both commercial and consumption purposes [9–11]. The increasing demand for cereal crops such as wheat, driven by population growth, necessitates strategic land use and management practices [12,13].

Crop production can be affected by different factors. Drought, in particular, has a more profound impact on human life compared to other natural disasters. The agricultural industry is especially vulnerable, facing significant challenges due to reduced rainfall [14]. In Ethiopia, drought conditions intensified over the past few decades, severely affecting agricultural productivity and socioeconomic stability. Several studies have highlights agricultural drought using multiple indices, such as the Vegetation Condition (VCI) and Vegetation Health Index (VHI), during crop growing season in Central Ethiopian Rift valley region, as well as northwestern and southwestern parts of Ethiopia [15–17]. Projections suggest that drought intensity and frequency will worsen, particularly in northern and northeastern regions, posing a grave threat to food security and livelihoods [18]. Addressing these challenges requires targeted intervention to enhance resilience and sustainable resource management. Consequently, agricultural drought is critical problem for food security in Ethiopia. To address these challenges, wheat production is considered a viable solution to improve food insecurity, increase national income, and self-sufficiency of food consumption [19,20]. Currently, the government Ethiopia has been implementing winter irrigation wheat production to address food insecurity and increase national income through the export of winter wheat, particularly in the eastern part of the country [21]. The government has determined the areas allocated for wheat production based on the land previously used for wheat production. The southwestern parts of the country, with their diverse agro-ecological conditions, offer substantial potential for expanding wheat farming. The Agro-Ecological Suitability approach identifies specific climate zones in the southwest as highly suitable for wheat production due to their sufficient rainfall and fertile soils [22,23].

Assessing land suitability for wheat cultivation is a crucial step in enhancing wheat production levels. In this approach, land is evaluated and categorized based on its capacity to sustain wheat growth, taking into account variables such as soil type, fertility, drainage, terrain, and climate. Identifying suitable locations ensures efficient resource allocation and focuses cultivation efforts on areas with the highest potential for yield [24–26]. Wheat Yield and land suitability are influenced by factors such as rainfall, temperature, moisture levels, nutrient availability, and the length of the growing season [27,28]. In the southwestern part of Ethiopia, various agricultural crops have been cultivated for an extended period. The major cereal crops produced in this region include coffee, maize, millet, sorghum, teff, and others. Despite the region's agricultural richness, wheat cultivation remains relatively limited compared to the eastern parts of the country. However, the southwestern region has significant potential for expanding wheat production due to its large areas of suitable land,

favorable climate, and diverse agroecological zones. By leveraging these natural advantages, the region could substantially enhance wheat productivity, contributing to food security and economic development in Ethiopia. Moreover, adopting improved farming techniques, advanced irrigation systems, and high-yield wheat varieties could further unlock this potential, positioning the southwestern part of the country as a key player in wheat cultivation.

The application of GIS and MCDA methods enables detailed spatial analysis and classification, offering a more integrated and robust assessment of land suitability for wheat cultivation. This approach surpasses the scope of previous studies. Moreover, despite previous studies, there is a lack of research on land suitability for *Triticum aestivum* L. in the southwestern parts of Ethiopia, particularly in the selected districts. The absence of evidence-based information has led to mismatches between cultivated land and wheat production, resulting in significant production losses. To address this issue and enhance wheat productivity, detailed scientific investigations were required. Geographical Information Systems (GIS) and remote sensing based MCDA are advanced technologies that help evaluate and identify suitable areas for specific land uses.

GIS allows for the collection, analysis, and interpretation of spatial data to map and assess land characteristics, while remote sensing involves using satellite images to monitor and analyze land conditions. Together, these technologies provide detailed and accurate information on factors such as soil properties, land uses, topography, infrastructures and climate, enabling more informed decisions about where different crops, including *Triticum aestivum* L., can be most effectively cultivated. Therefore, the present study aims to address the existing research gap by assessing potential land suitability for wheat production by using GIS and multi criteria decision analysis in southwestern parts of Ethiopia.

## 2. Materials and methods

### 2.1. Descriptions of the study area

The research was conducted in selected districts in southwestern parts of Ethiopia. The study area contained 22 districts which are found in Jimma zone, Buno Bedele zone and West Wollega zone. Geographically, the study area is situated between 8°20′00″to 9°10′00″N and 35°11′30″ to 36°53′30″E (Fig 1). Topographic variation of the selected districts located from 952m to 2580m above the sea level. Total coverage of the study area is 14073.8km$^2$. The climate of southwestern parts of Ethiopia presents both challenges and opportunities for wheat production. This region experiences significant variability in rainfall and temperature, which directly impacts agricultural practices and crop yields. Understanding these climatic conditions is crucial for optimizing wheat production strategies. Southwestern parts of Ethiopia particularly the study area, receive maximum rainfall during the summer season (June to September), ranging from 1445 to 2246 mm. The maximum and minimum temperature in the study area varies between 15.4°C and 34.2 °C, respectively.

### 2.2. Data sources and descriptions

For assessment of potential land suitability for rainfed wheat production in the selected districts, different biophysical data were applied. These data were climate data, Landsat data, topographic data, soil data and infrastructure data. Specifically, rainfall data of the study area was collected from national metrological institute by rainfall metrological stations which located in and around of the study area. Landsat OLI/TIRS of 2024 was downloaded from earth explorer website to calculate land surface temperature and to classify land use land cover types of the study area. Digital elevation model was downloaded from United States Geological Survey (USGS) website to generate elevation and slope of the selected districts. Soil data like soil drainage, soil texture and soil depth were obtained from the Ministry of Water and Energy. Finally, road and towns of the study area were obtained from Ethiopian Mapping Agency (Table 1). These parameters were selected based on substantial impacts on potential land suitability for wheat production in the study area.

To analyze the potential land suitability for wheat production of the study area, several softwares were applied. These softwares were, ArcGIS 10.3, ERDAS Imagine 2015, ArcSWAT, and IDRISI Selva 17.

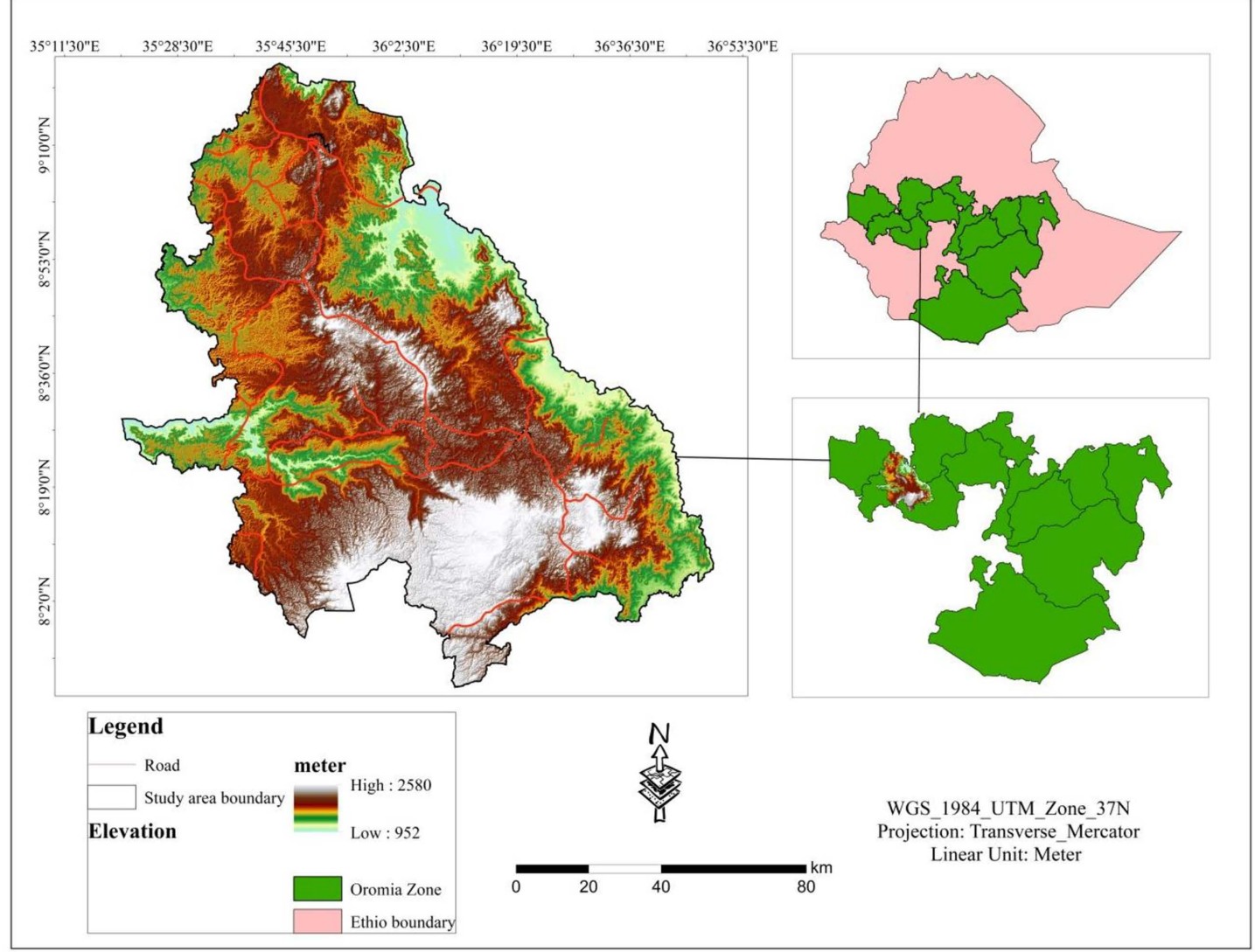

**Fig 1. Location map of the study area.**

**Table 1. Types and uses of data.**

| Data | Data types | Resolution(m) | Sources |
|---|---|---|---|
| Climate data | Rainfall data | 30 | National Meteorological Institute |
| Landsat OLI/TIRS | Land Surface Temperature, Land use land cover | 30 | https://earthexplorer.usgs.gov/ |
| STRM DEM | Elevation, Slope | 30 | https://earthexplorer.usgs.gov/ |
| Soil data | Soil drainage, soil depth, soil texture | 30 | Ministry of Water and Energy |
| Infrastructure data | Roads and Market | 30 | Ethiopian mapping Agency |

## 2.3. Data analysis

By using GIS-based multi-criteria evaluation, ten parameters were aggregated to evaluate potential land suitability for rainfed wheat cultivation in the western parts of Ethiopia. To convert vector data to raster form, Rasterization was performed for appropriate weighted overlay analysis. Weighted importance value of each factors were calculated by using Analytical Hierarchy Process (AHP) method in IDRISI Selva software environment. After resolution of all factors resample to 30m, it reclassified into five classes (highly suitable, moderately suitable, marginally suitable, currently not suitable and permanently not suitable). According to [29] stated that, pairwise comparison matrix was made to weight and relative importance of each parameter by normalized the egnivector of the factors by their cumulative total (Table 2).

## 2.4. Physical land suitability for wheat cultivation

**2.4.1 .Slope.** Slope gradient is one of the most important factors for assessing potential land suitability for rainfed wheat cultivation. According to [30], the land with a gentle or flat slope is more suitable than a steep for wheat cultivation. Steep slopes are highly vulnerable to soil erosion which can significantly reduce yield production. The slope of the study area was classified into five classes in percentages, reflecting their suitability for wheat cultivation. These categories are: 0–13%, 13–25%, 25–40%, 40–55% and greater than 55% as highly suitable, moderately suitable, marginally suitable, currently not suitable and permanently not suitable respectively [31,32].

**2.4.2. Land surface temperature (LST).** Climate data, particularly temperature, is an essential factor for rainfed wheat cultivation. According to previous studies [31], cooler temperatures are more suitable for wheat cultivation than warmer temperatures. As the result, temperature was reclassified into the following categories: $<22^{\circ}C$, 22 to $25^{\circ}C$, 25 to $27^{\circ}C$, 27 to $29^{\circ}C$ and greater than $29^{\circ}C$ as highly suitable, moderately suitable, marginally suitable, currently not suitable and permanently not suitable respectively.

**2.4.3. Elevation.** Altitude was another environmental factor used to assess the suitability of wheat cultivation of wheat in the study area. Based on its suitability, altitude was classified into the following ranges: $>2100m$, 1800 to 2100m, 1600to1800m,1400 to 1600m, 950 to 1400m. The highland area was more suitable than low land area for wheat crop cultivation in the study area. According to [33], altitudes between 1900 and 2700 meters above sea level are most suitable for wheat production due to their cooler temperatures and high rainfall conditions.

**Table 2. Rating parameters for Wheat land suitability analysis.**

| Parameters | Classification criteria and scale | | | | |
|---|---|---|---|---|---|
| | Unit | Highly suitable | Moderately suitable | Marginally suitable | Currently not suitable | Permanently not suitable |
| LULC | Class | cultivated land | Grassland | Shrubs land | bare land | Forest land water body settlement |
| Soil drainage | Class | Well | Moderate | Somewhat excessive | Excessive | Imperfect |
| Soil texture | class | Sandy clay | clay Loam, Loam | Sandy clay loam | Sandy loam | Loamy sand |
| Soil depth | cm | >100 | 75-100 | 50-75 | 50−25 | <25 |
| Market | km | 0–4 | 4–8 | 8–12 | 12-18 | >18 |
| Road | km | 0–3 | 3–7 | 7–12 | 12-18 | >18 |
| LST | °C | <22 | 22-25 | 25-27 | 27-29 | >29 |
| Rainfall | mm | >1916 | 1836-1916 | 1768-1836 | 1695-1768 | <1695 |
| Slope | % | 0-13 | 13-25 | 25-40 | 40-55 | >55 |
| Elevation | m | >2100 | 1800-2100 | 1600-1800 | 1400-1600 | 950-1400 |

**2.4.4. Soil depth.** Soil depth of the study area was extracted from Ethiopian soil depth data using the study area boundary. Soil depth indicates the volume and root space of the soil, which determines the availability of water and nutrients for crops. This factor influences the types of crops cultivated in the study area, as deeper soils generally provide more nutrients to crops compared to shallow soils. According to [31], soil depth of the study area was reclassified from high deep to shallow based on its suitability for wheat cultivation. As the result, soil depths were categorized as follows: greater 100 cm, 75–100 cm, 50–75 cm, 50−25 cm and less than 25 cm were highly suitable, moderately suitable, marginally suitable, currently not suitable and permanently not suitable respectively.

**2.4.5. Land use land cover (LULC).** Land use and land cover types of the study area were classified using Landsat OLI/TIRS imagery from 2024 through supervised classification with the maximum likelihood algorithm. The LULC types were categorized into cultivated land, grassland, shrubs land, bare land, settlement, forest land and water bodies. Among the classified LULC types, cultivated land, grassland, shrubs land and bare land were highly suitable, moderately suitable, marginally suitable, and currently not suitable respectively, for wheat crop production. In contrast, settlement, forest land, and water bodies were classified as permanently not suitable for wheat production in the study area [31].

**2.4.6. Soil drainage.** Soil drainage refers to the upward or downward movement of water through the soil profile. It plays a crucial role in reducing soil nutrient loss caused by runoff and soil erosion in the study area. For wheat production, soil drainage is essential as it allows excess water to drain from the field, promoting better crop growth [34]. The study area contains five drainage classes: well-drained, moderately drained, somewhat excessively drained, and imperfect drained. These classes were classified as highly suitable, moderately suitable, marginally suitable, currently not suitable and permanently not suitable, respectively, for rainfed wheat production.

**2.4.7. Soil texture.** Soil texture refers to the proportion of soil particles, such as sand, silt, clay and loam, which influence mineral composition of the soil and its suitability for crop production [35]. Different crops prefer different soil textures for optimal germination and growth. Therefore, soil texture significantly affects the potential land suitability for rainfed wheat production in the study area. The soil texture data was then imported into GIS software, ensuring it matched the spatial reference system of the study area. A polygon defining the boundaries of the study area was created to focus on the relevant region. Using this polygon, the soil texture data was clipped to isolate the information within the study area. The extracted data was then verified for accuracy, potentially comparing it with field data or original datasets. Finally, the soil texture data was analyzed to assess soil properties within the study area, and the extraction process was documented to ensure clarity and reproducibility. According to the [31], the soil textures were classified as follows: sand clay (highly suitable), loam and clay loam (moderately suitable), sandy clay loam (marginally suitable), sandy loam (currently not suitable) and loamy sand (permanently not suitable) for wheat cultivation in the study area.

**2.4.8. Rainfall.** Rainfall data, obtained from stations managed by Ethiopian Metrological Institute, was imported into GIS software, ensuring proper alignment with the geographic boundaries of the study area. The average, minimum, and maximum rainfall amounts were calculated over various time intervals. To visualize the spatial distribution of rainfall, interpolation techniques were employed. Specifically, the Inverse Distance Weighting (IDW) method was used for interpolation. This technique estimates rainfall values at unsampled locations based on the values of nearby data points, with closer points having a greater influence on the estimated values. Maps depicting the spatial distribution of rainfall were generated using this method, providing a clear visualization of rainfall patterns across the study area. Established classification criteria were applied to categorize the rainfall amounts into suitability classes for wheat cultivation rainfall [31]. The areas were classified into the following suitability categories based on rainfall data: highly suitable (greater than 1916mm), moderately suitable (1836–1916mm), marginally suitable (1768–1836mm), currently not suitable (1695–1768 mm) and permanently not suitable (less than 1695 mm). The classification results were validated by comparing them with historical data and, where feasible, corroborating with field observations.

**2.4.9. Proximity to market.** Distance from market was calculated using the spatial analyst tool in ArcGIS environment [36,37]. The Potential land suitability of the study area is influenced by proximity to markets, with areas closer to markets

being more favorable than those farther away. Farmland located far from the markets often faces challenges such as market access and reduced supply for production. In contrast, areas closer to the markets are characterized by transportation costs, reduced transaction costs and fewer other marketing-related expenses. According to [38], distances were classified into the following suitability categories for rainfed wheat cultivation in the southwestern parts of Ethiopia: 0–4 km, 4–8 km, 8–12 km, 12–18 km and greater than 18 km were highly suitable, moderately suitable, marginally suitable, currently suitable and permanently not suitable respectively.

**2.4.10. Proximity to road.** Road accessibility enables farm owners to transport their yields from the farmland to the market center and bring necessary input to the farming site without difficulty [39,40]. Therefore, land suitability for wheat cultivation depends on proximity to the roads. Areas closer to the roads are most suitable, while those farther away are less suitable for rainfed wheat cultivation in the study area. Proximity to roads in the study area was reclassified into 0–3 km, 3–7 km, 7–12 km, 12–18 km and greater than 18 km and corresponding to highly suitable, moderately suitable, marginally suitable, currently not suitable and permanently not suitable respectively.

### 2.5 .Multi-criteria evaluation (MCE)

Multi-criteria Evaluation (MCE) is a widely utilized technique for assessing potential land suitability for rainfed wheat production. It integrates various geospatial datasets and criteria to assess and identify suitable zones for wheat cultivation. The approach typically involves combining thematic layers such as slope, LST, elevation, soil depth, soil texture, soil drainage, rainfall, LULC, proximity to market and proximity to roads. Each layer is weighted based on its influence on potential land suitability for wheat cultivation. Analytical Hierarchy Process (AHP) is often employed to assign weights to these criteria through pairwise comparisons, ensuring the a systematic and rational decision-making process [41,42]. The Multi-criteria Evaluation (MCE) approach, utilizing the Analytical Hierarchy Process (AHP), was applied to calculate the criteria weights for assessing land suitability for rainfed wheat cultivation. This method follows the 1-to-9 scale proposed by [29], which enables pairwise comparisons of selected parameters to evaluate their relative importance systematically. These comparisons facilitate the reclassification and weighting of criteria based on their significance and influence on potential land suitability for selected crops in the study area. The Analytical Hierarchy Process (AHP) is a valuable tool for decision making and it depends on the specific problem and quality of the data used. It is known for its high accuracy compared to other methods [43].

The comparison of parameters was assigned based on expert opinion and previously published papers related to the potential land suitability of wheat production. The relative importance of the parameters including, slope, LST, elevation, soil depth, soil texture, soil drainage, rainfall, LULC, proximity to market and proximity to road were calculated using IDRISI Selva 17 software (Table 3). The consistency and clarity of the pairwise comparisons were assessed using the consistency ratio (CR), with an acceptable CR value set at less than 10%. The CR was determined as the ratio of the consistency index (CI) to the random consistency index, ensuring reliability in the weight assignment process. This systematic approach enhances the robustness of potential land suitability for wheat cultivation assessment (1).

$$CR = \frac{CI}{RI} \tag{1}$$

where CI is the consistency index and RI is the random consistency index.

Consistency index is the measure of parameters consistency as the degree of consistency by using (2):

$$CI = \frac{\lambda \max - n}{n-1} \tag{2}$$

where n indicates the number of parameters, $\lambda\, max$ refers to is the principal Eigenvalue of parameters that can be calculated from the multiplication of the total horizontal summation of given intensity importance value and parameters

**Table 3. pair wise comparison matrix of selected parameters.**

| Factors | Slope | Elevation | LST | Soil depth | LULC | Soil drainage | Soil texture | Rainfall | Market | Road | Weight |
|---|---|---|---|---|---|---|---|---|---|---|---|
| Slope | 1 | 2 | 2 | 2 | 3 | 3 | 3 | 4 | 4 | 5 | 0.18 |
| Elevation | ½ | 1 | 2 | 2 | 2 | 3 | 3 | 4 | 4 | 5 | 0.17 |
| LST | ½ | 1/2 | 1 | 2 | 2 | 2 | 3 | 3 | 3 | 4 | 0.13 |
| Soil depth | ½ | 1/2 | 1/2 | 1 | 2 | 2 | 3 | 3 | 3 | 4 | 0.12 |
| LULC | 1/3 | 1/2 | 1/2 | ½ | 1 | 2 | 3 | 3 | 3 | 3 | 0.11 |
| Soil drainage | 1/3 | 1/3 | 1/2 | ½ | ½ | 1 | 2 | 3 | 3 | 4 | 0.1 |
| Soil texture | 1/3 | 1/3 | 1/3 | 1/3 | 1/3 | ½ | 1 | 2 | 3 | 3 | 0.08 |
| Rainfall | ¼ | 1/4 | 1/3 | 1/3 | 1/3 | 1/3 | ½ | 1 | 2 | 3 | 0.06 |
| Market | ¼ | 1/4 | 1/3 | 1/3 | 1/3 | 1/3 | 1/3 | ½ | 1 | 2 | 0.03 |
| Road | 1/5 | 1/5 | 1/4 | ¼ | 1/3 | ¼ | 1/3 | 1/3 | ½ | 1 | 0.02 |
| Σ | 5.2 | 5.87 | 7.7 | 9.2 | 11.83 | 14.36 | 19.16 | 23.83 | 26.5 | 34 | 1 |

$\lambda$ max= (5.2*0.18) +(5.87*0.17)+ (34*0.02)=11.2138, n=10, CI=0.134, RI=1.49, CR=0.08.

normalized principal Eigenvectors value. The normalized principal Eigenvector was obtained by averaging the normalized relative weight of the parameters.

The random consistency index is the constant number assigned for each targeted parameter based on their intensity importance scale (Table 4).

### 2.6. Potential land suitability analysis for wheat production

Weighted overlay analysis was applied to assess potential land suitability for wheat production in the study area and it calculated in (3)

$$S = \Sigma Wi * Xi \tag{3}$$

Where: Wi is weight of factor and Xi is criterion score of factors i

### 2.7. Inclusivity in global research

Additional information regarding the ethical, cultural, and scientific considerations specific to inclusivity in global research is included in the supporting information.

## 3. Results and discussions

### 3.1. Slope

Slope gradient of the study area was generated in percentages to assess potential land suitability for wheat cultivation in the southwestern parts of Ethiopia. Most parts of the study area, particularly eastern and central regions, were flat and had gentle slopes, making them more favorable for wheat cultivation to other areas (Fig 2a). This finding aligned with the findings of [44,45] flat slopes or less slopes of land were suitable for wheat cultivation in their study

**Table 4. Random Index Value.**

| Intensity importance | 1 | 2 | 3 | 4 | 5 | 6 | 7 | 8 | 9 | 10 |
|---|---|---|---|---|---|---|---|---|---|---|
| Constant importance | 0.00 | 0.00 | 0.58 | 0.90 | 1.12 | 1 | 1.32 | 1.41 | 1.45 | 1.49 |

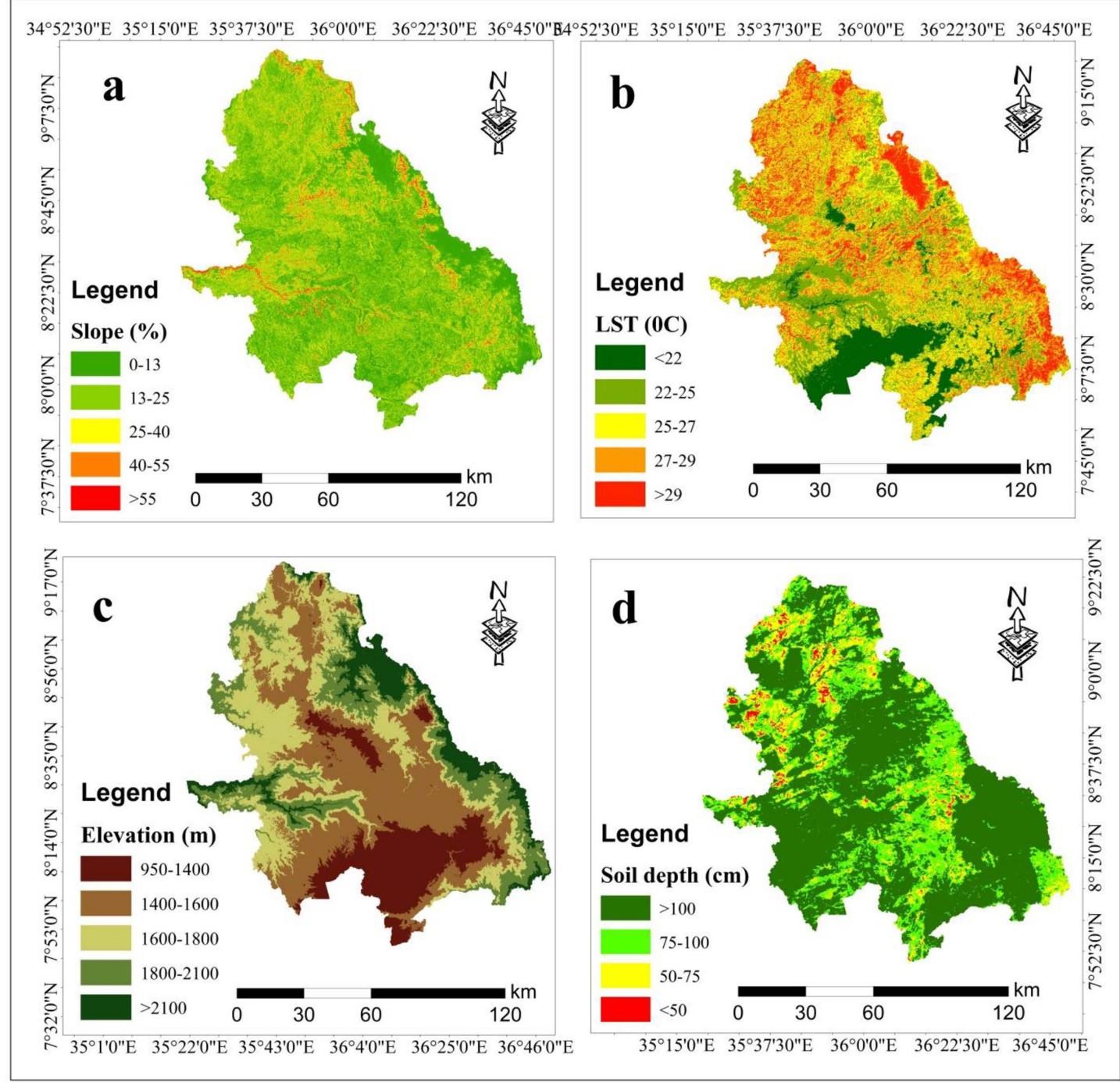

**Fig 2. Slope** (a), LST (b), Elevation (c) and Soil depth (d).

area. Sloped land tends to experience more surface runoff, as water flows downhill quickly, carrying away valuable topsoil and nutrients. Cultivating wheat in these areas helps slow down the flow of water, as the roots of wheat plants create channels that allow water to infiltrate the soil rather than flowing off the surface. The canopy of the wheat also intercepts rainfall, reducing the direct impact of raindrops on the soil, which further reduces erosion. By integrating wheat cultivation into agricultural practices, farmers can boost soil fertility, paving the way for sustainable and productive farming systems [46,47].

### 3.2. Land surface temperature (LST)

The analysis of land surface temperature reveals that areas with cooler temperatures are more suitable for wheat cultivation in the study area compared to those with warmer temperatures. Spatially, the southern and western parts of the study area were more suitable than northern and eastern parts of the study area (Fig 2b). This distinction is likely due to variations in climatic conditions, such as cooler temperatures and potentially higher soil moisture retention in the southern and western part of the study area. The optimal time to plant wheat largely depends on temperature, as it plays a critical role in the crops' growth, development, and eventually grain yield. Wheat thrives in cool weather during its vegetative growth phase, making it a crop best suited for regions with mild winters. Previous studies have concluded that heat stress occurs when temperatures exceed 25°C during critical growth stages, such as flowering and grain filling, in wheat production [28,48–50].

### 3.3. Elevation

Elevation plays a critical role in shaping the temperature and rainfall patterns of the study area. Geographically, eastern and some western parts of the study area were highly suitable for rainfed wheat cultivation compared to central and southern parts of the study area (Fig 2c). Previous studies have demonstrated that the highland areas are highly suitable for wheat production, while midland areas are moderately suitable. In contrast, lowland areas are generally not considered suitable [51]. Some scholars have highlighted that highland areas play a crucial role in enhancing the overall suitability of the country for wheat production under current conditions [28,52]. Meanwhile, other regions of the country are considered moderately suitable for wheat cultivation, provided that appropriate management practices are applied [53].

### 3.4. Soil depth

The type of wheat and the soil conditions determine the optimal depth for its growth. Sufficient sunlight and good drainage are essential for successful cultivation. Soil depth, which refers to the depth at which soils are located, can influence the rate at which water and essential nutrients are absorbed by the soil from the earth's crust. In the study area, soil depths exceeding 100 cm are well drained and optimal for growing wheat. Spatially, central and southern parts of the study area were highly suitable than northern parts of the study area for rainfed wheat production (Fig 2d).

The study's findings align with those of [54], who categorized very high depth (100–150 cm) as optimal for wheat crop production. The conclusion that deeper soils more favorable than shallow soils for wheat cultivation is consistent with previous research conducted regions of Ethiopia, including the southeastern Ethiopian highlands, the southern highlands, and western Ethiopia [55,56]. These studies have consistently highlighted the importance of soil depth in determining the success of wheat farming, suggesting that deeper soils provide a more conducive environment for wheat growth compared to shallow soils. Additionally, the finding of this study was supported by the research conducted by [45], which indicated that approximately 71.8% of the watershed in their study was classified as highly suitable for wheat cultivation based on soil depth. Furthermore,15% of the area was deemed marginally suitable, while 13% were considered unsuitable for wheat production.

### 3.5. Land use land cover (LULC) types

There are seven LULC classes in the study area: forest land, cultivated land, bare land, grassland, shrubs land, settlement and water body. According to [31], cultivated land, grass land and shrubs land were highly as highly suitable, moderately suitable and marginally suitable respectively for wheat crop production, respectively, for wheat crop production. However, bare land, forest land, settlement and water body were categorized as currently not suitable and permanently not suitable for rainfed wheat production in the southwestern parts of Ethiopia.

The results revealed that about 56.3% of LULC types were highly suitable whereas 4.5% of LULC types were moderately suitable land for wheat cultivation. Additionally, about 15.6% of the study area was marginally suitable, whereas 0.5% and 23.2% of the study area were classified as currently not suitable and permanently not suitable, respectively, for wheat cultivation in the study area (Table 5). Geographically, the eastern and some central parts of the study area were occupied with highly suitable while some parts of central and southern parts of the study area were dominated with moderately suitable land cover types. Similarly, the remaining parts of western and some southern areas of study area were captured with marginally suitable, currently not suitable and permanently not suitable for wheat production (Fig 3a). The finding of this research was consistent with results of a study by [57]; which found that 74% their study area was highly suitable and 24% was moderately suitable for wheat cultivation in the study area.

### 3.6. Soil drainage

The speed of water movement in the soil profile depends on the size of soil pores present in the soil type. Soil with larger pore sizes has higher soil drainage capabilities. Based on these physical features, the study area was classified in to well drained, moderately drained, somewhat excessively drained, excessively drained, and imperfectly drained categories. According to the previous scholar's reports, the soil types characterized with well and moderate soil drainage features were considered as highly suitable and moderately suitable, while soil with somewhat excessive, excessive and imperfect soil drainage capacity were ranked as marginally suitable, currently not suitable and permanently not suitable respectively for wheat cultivation [58–60]. The results show about 83.43% of soil drainage of the study area was highly suitable for wheat cultivation. Additionally, about 2.09% and 13.09% of the study area were occupied by moderately suitable and marginally suitable, respectively, for wheat crop production (Table 6). Spatially, most parts of the study area were dominated by highly suitable soil drainage, while some northern and southern parts of study area were occupied by moderately suitable and marginally suitable soil drainage (Fig 3b).

### 3.7. Soil texture

In Ethiopia, loamy soil a mixture of sand, silt, and clay that is well-drained and nutrient rich is ideal for growing wheat. Wheat thrives in loamy soil because it promotes healthy air circulation and root development. In the study area, sand clay

**Table 5. LULC classes of study area.**

| LULC Types | Area (km²) | Area (%) |
|---|---|---|
| Bare land | 67.0 | 0.5 |
| Cultivated land | 7919.8 | 56.3 |
| Forest land | 3093.4 | 22.0 |
| Grassland | 631.3 | 4.5 |
| Settlement | 88.7 | 0.6 |
| Shrubs land | 2198.3 | 15.6 |
| Water body | 78.4 | 0.6 |
| Total | **14076.8** | **100** |

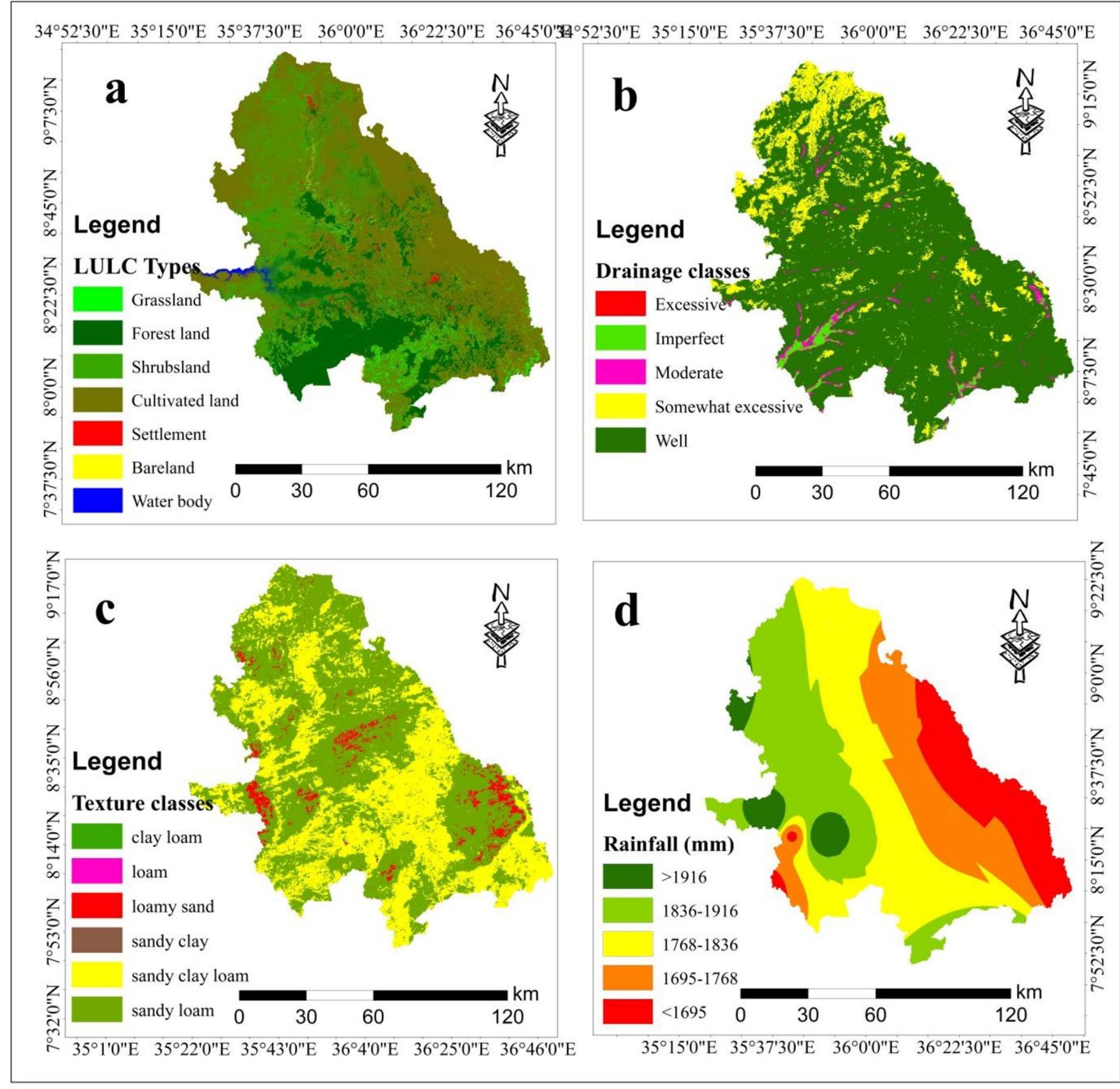

**Fig 3. LULC** (a), Soil drainage (b), soil texture (c) and Rainfall (d).

**Table 6. Soil drainage classes of study area.**

| Soil drainage | Area (km²) | Area (%) |
|---|---|---|
| Imperfect | 192.5 | 1.37 |
| Moderate | 294.5 | 2.09 |
| Well | 11741.7 | 83.43 |
| Somewhat excessive | 1842.6 | 13.09 |
| Excessive | 2.5 | 0.02 |
| Total | **14073.8** | **100.00** |

**Table 7. Soil texture classes of study area.**

| Soil Texture | Area (km²) | Area (%) |
|---|---|---|
| sandy clay | 10.5 | 0.1 |
| clay loam | 11.8 | 0.1 |
| sandy clay loam | 5478.0 | 38.9 |
| Loam | 14.3 | 0.1 |
| sandy loam | 8082.3 | 57.4 |
| loamy sand | 476.9 | 3.4 |
| Total | **14073.8** | **100** |

was classified as highly suitable, while loam and clay loam were moderately suitable for wheat cultivation. Sandy clay loam, on the other hand, was categorized as marginally suitable for rainfed wheat cultivation in the study area [22].

The results show that about 0.1% and 0.2% of the study area were occupied as highly suitable and moderately suitable, respectively, in terms of soil texture for wheat crop production. In contrast, about 38.9% and 57.4% of the study area were covered by marginally suitable and currently not suitable soil textures, respectively, for wheat production (Table 7). Geographically, the central, western, and eastern parts of study area were dominated by marginally and currently not suitable soil textures for wheat cultivation (Fig 3c). The result was in line with the finding of [61,62], as they categorized the types of soil silt clay, silt loamy silt, sandy clay-loam and clay. Silt loam accounts for dominant soil type and is highly suitable for wheat cultivation in their study area.

### 3.8. Rainfall

The productivity of the area was assessed by examining the availability and timing of rainfall, which plays a critical role in ensuring optimal wheat growth [63,64]. This evaluation focused on determining whether the rainfall patterns were sufficient and well-timed to support the different growth stages of the wheat crop, including germination, tillering, flowering, and grain filling. By correlating these factors with the yield per unit area over a specific period, the study aimed to understand how effectively natural rainfall met the crop's water requirement and influenced overall agricultural productivity in the region. According to [31], stated that, the area dominated with high rainfall was highly suitable than low rainfall for rainfed wheat cultivation in the southwestern parts of Ethiopia. Spatially, the western part of the study area was highly suitable whereas, eastern and central parts of the study area were marginally suitable and currently not suitable respectively for wheat cultivation in the study area (Fig 3d). This finding was consistent with the finding of [61] who summarized their finding as the highest rainfall area being the most suitable for wheat cultivation. The result also found by [65,66], rainfall is a major factor of wheat yield influencing everything from seed germination to grain filling. Both the quantity and timing of rainfall are crucial with any disruption in water availability at critical growth stages potentially leading to lower yields and

poor grain quality. Farmers, therefore, must closely monitor weather patterns and adopt strategies that optimize water use to ensure the best possible wheat production outcomes.

## 3.9. Proximity to market

Market suitability is critical for the success of wheat cultivation, as it directly affects the economic viability of farming operations. Proximity to well-established markets plays a pivotal role in ensuring farmers to sell their wheat at competitive prices and obtain fair returns. It reduces transportation costs, improves access to a wider range of buyers, enhances bargaining power, and allows farmers to diversify their sales channels. By facilitating better financial and technical support, ensuring price stability, and providing access to export opportunities, proximity to markets can significantly enhance the profitability and sustainability of wheat farming [67,68]. Moreover, efficient market access minimizes post-harvest losses by facilitating timely sales, thereby ensuring the produce remains fresh and high quality. This enhances overall supply chain efficiency and supports the financial security of wheat farmers, enabling them to reinvest in improved farming practices and technologies [69]. In this context, greater proximity to the markets implies higher suitability for farming activates. The results revealed that the market distance analysis of the study area ranged from 0 to 4 km, 4–8 km, 8–12 km, 12–18 km and greater than 18 km with a suitability class of highly suitable, moderately suitable, marginally suitable, currently not suitable and permanently not suitable, respectively, for wheat cultivation. Geographically, the southeastern, southwestern and northern areas of study area were classified as highly suitable and moderately suitable, while some central and southern parts were categorized as marginally suitable and currently not suitable for wheat production (Fig 4a).

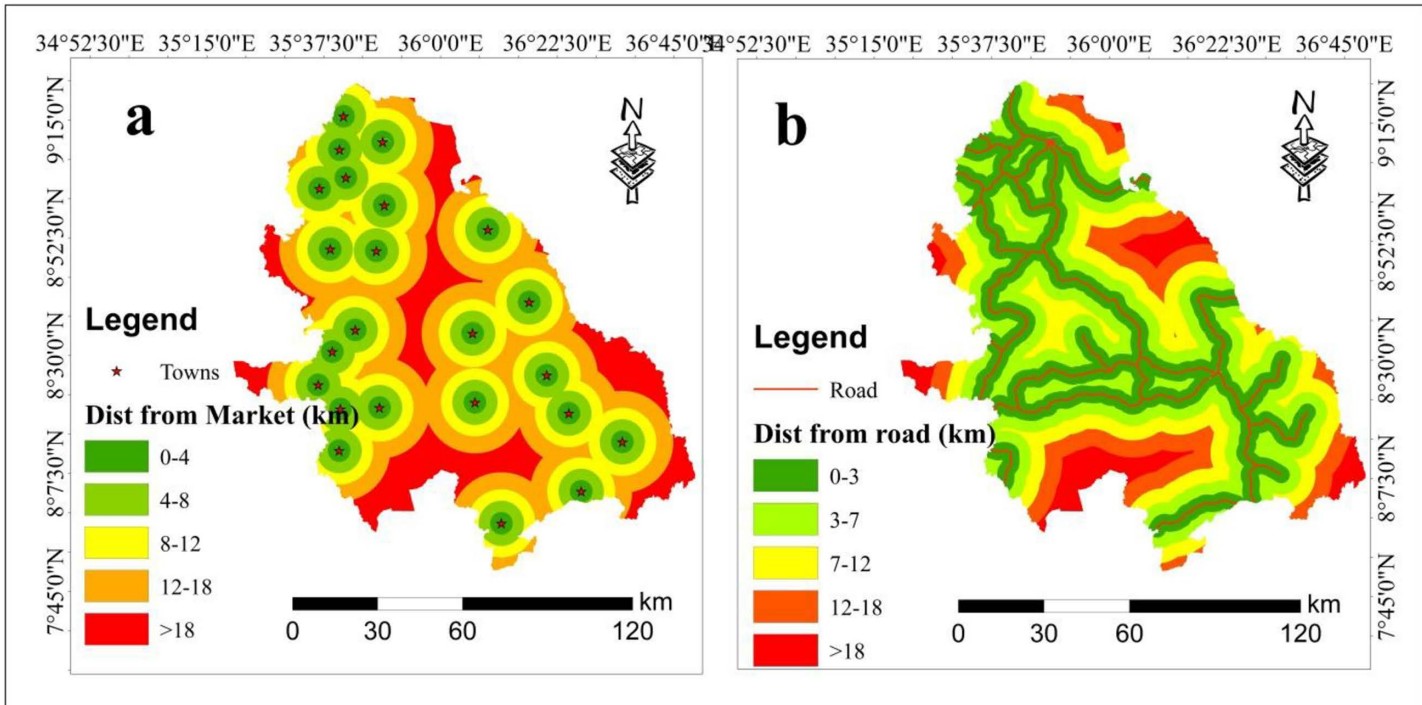

**Fig 4. Distance from market** (a) and Distance from road (b).

## 3.10. Proximity to road

Road infrastructure plays a crucial role in determining the suitability of land for wheat cultivation, as it directly impacts transportation efficiency and costs. Well-maintained roads reduce both travel time and transportation expenses, enabling farmers to transport their wheat from farms to markets more efficiently. This not only lowers production costs by reducing fuel and vehicle maintenance expenses but also enhances farmers' access to larger and more profitable markets, boosting both local and regional economies. Additionally, good road infrastructure allows farmers to reach more diverse customer bases, which is essential for scaling up production and enhancing market reach [38,70,71].

Additionally, effective road infrastructure supports timely emergency responses and the distribution of resources, further contributing to the overall success and sustainability of wheat farming [69]. Areas farther from roads are less suitable for farming compared to those closer to roads. acceptability than the nearest one. In (Fig 4b), the study area was classified into different land suitability categories based on proximity to roads: highly suitable, moderately suitable, marginally suitable, currently not suitable and permanently not suitable based on proximity of the road 0–3 km, 3–7 km, 7–12 km, 12–18 km and greater than 18 km respectively. Spatially, the central and southern parts of the study area were dominated by highly suitable land, and moderately suitable while the northern and some western parts were classified as marginally suitable and currently not suitable land for rainfed wheat cultivation in the southwestern parts of Ethiopia.

## 3.11. Potential land suitability evaluation for wheat (*Triticum aestivum L*) production in Southwestern parts of Ethiopia

The aggregation of infrastructural, ecological, and biophysical data within a Geographical Information System environment plays a crucial role for evaluating the suitability classes of the study area for achieving high yields of rainfed wheat crop. The result revealed that about 1.3% of the study area was classified as highly suitable for wheat crop cultivation. Additionally, about 38.2% of the study area was categorized as moderately suitable for wheat cultivation. However, about 51.5% and 8.8% of the study area were characterized as marginally suitable and currently not suitable, respectively, for rainfed wheat production in southwestern parts of the study area (Table 8). High suitable land classes resulted from the integrations of favorable factors, including gentle slopes, higher rainfall, optimal elevation, low temperature, proximity to markets and roads, cultivated land cover types, and suitable soil characteristics within the study area. Conversely, the absence or reversal of these factors contributes to the unsuitability of the study area for wheat crop cultivation.

Highly suitable land classes situated to the eastern and southern parts whereas the specific location of moderately suitable land classes was north east, North West and south east of the study area. In addition, marginally suitable land classes occupy the central parts while unsuitable land classes cover some southern parts of the study area (Fig 5). This finding is in line with the previous studies [22,31], which conducted in western and eastern Ethiopian highlands. In addition, identifying potential land suitability and planning land use for sustainable land management and cereal crops production using GIS and remote sensing is essential for decision makers [72–76].

**Table 8. Suitability classes and area coverage for wheat crop production.**

| Suitability classes | Area (km2) | Area (%) |
|---|---|---|
| Highly suitable | 177.1 | 1.3 |
| Moderately suitable | 5375.2 | 38.2 |
| Marginally suitable | 7246.0 | 51.5 |
| Currently not suitable | 1235.1 | 8.8 |
| Permanently not suitable | 40.3 | 0.3 |
| Total | **14073.8** | **100.0** |

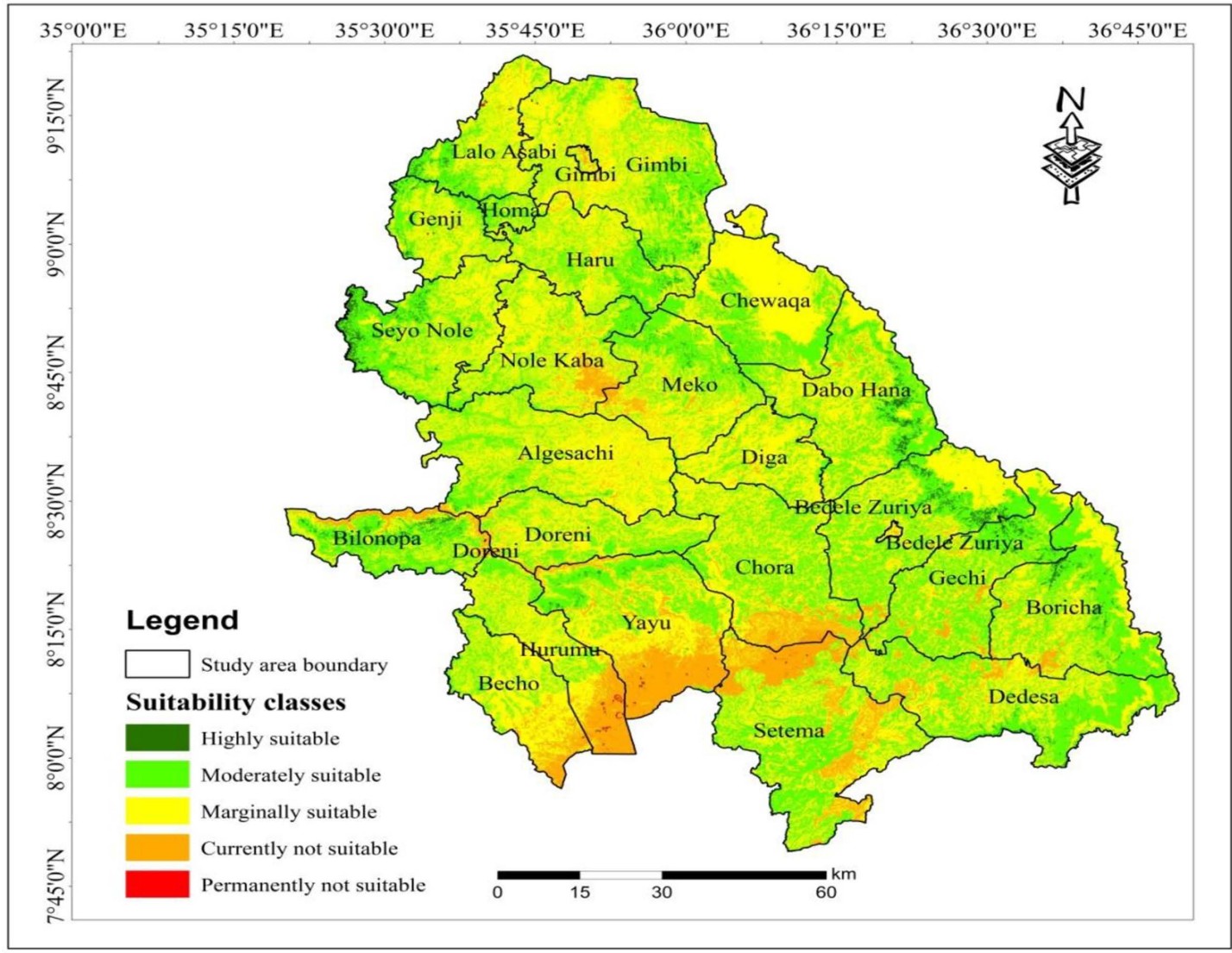

**Fig 5. Potential land suitability for Wheat crop production in the study area.**

### 3.12. Potential land suitability for wheat *(Triticum aestivum L)* cultivation across the districts in the southwestern part of Ethiopia

To make our study more comprehensive, we have separately assessed the land suitability of southwestern part of Ethiopia for wheat cultivation. Identifying land suitability is crucial for increasing agricultural productivity and ensuring that the most appropriate areas are prioritized for wheat farming. This approach is essential for establishing the region as a key zone for wheat cultivation, particularly given Ethiopia's growing demand for wheat and its challenges in food security and agricultural sustainability.

As depicted in the (Fig 6, Table 9), districts such as Sayo Nole (32.7km²), Bedele Zuriya (23.3km²), Daba Hana (21.5km²), Bilo Nopa (19km²) and Boracha (13.2km²) are currently identified as highly suitable areas for wheat cultivation

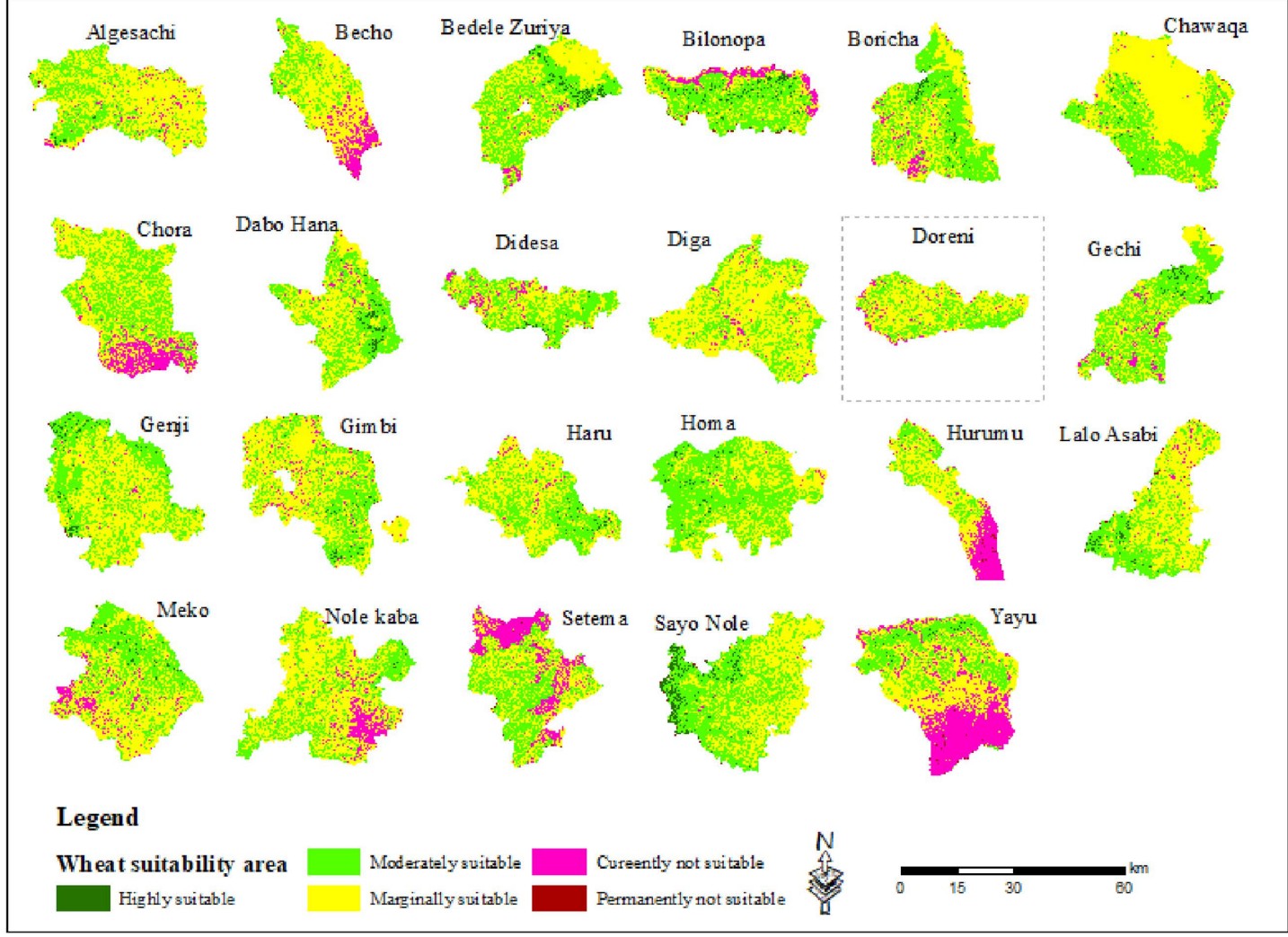

**Fig 6. Land suitability for wheat crop production across the districts in the study area.**

in the southwestern part of Ethiopia. The districts like Diga (0km²), Becho (0.1 km²), Setema (0.2 km²), Chora (0.3 km²) and Hurumu (0.5 km²) were currently not highly suitable for wheat cultivation. When comparing Moderate suitability, Didesa district (394.9 km²) stands out as one of the most moderately suitable areas for wheat cultivation within the study area. In contrast, Homa district (47.4 km²) is among the least moderately suitable, ranking low in comparison to the other 22 districts in the study area.

As shown in Table 9, when comparing of Marginal suitability across the districts in the study area, Algesachi district (577.7 km²) emerges as one of the most marginally suitable for wheat cultivation. On the other hand, Homa district (46.7 km²) is among the least marginally suitable for wheat cultivation when compared to the other districts in the study area. When comparing the districts that are currently unsuitable for wheat cultivation, Setema district (237.53 km²) stands out as one of the least unsuitable areas in the study area. Additionally, Homa district (0 km²) is one of the most permanently not suitable for wheat cultivation in the southwestern part of Ethiopia.

**Table 9. Suitability classes and area coverage for wheat crop production by each district.**

| No | District Vs Suitability area (km²) | Highly suitable | Moderately suitable | Marginally suitable | Currently not suitable | Permanently not suitable | Total |
|---|---|---|---|---|---|---|---|
| 1 | Algesachi | 1.7 | 260.2 | 577.7 | 33.9 | 1.5 | 875.0 |
| 2 | Becho | 0.1 | 98.5 | 286.2 | 77.1 | 2.6 | 464.5 |
| 3 | Bedele Zuriya | 23.3 | 339.3 | 362.6 | 18.4 | 1.7 | 745.3 |
| 4 | Bilonopa | 19.0 | 208.9 | 122.9 | 43.1 | 2.7 | 396.7 |
| 5 | Boricha | 13.2 | 363.7 | 329.6 | 31.1 | 1.9 | 739.5 |
| 6 | Chewaqa | 3.6 | 204.7 | 403.7 | 5.9 | 1.0 | 619.0 |
| 7 | Chora | 0.3 | 283.4 | 393.9 | 110.4 | 0.6 | 788.7 |
| 8 | Dabo Hana | 21.5 | 339.5 | 369.3 | 15.6 | 1.3 | 747.2 |
| 9 | Didesa | 8.1 | 394.9 | 381.2 | 55.2 | 2.6 | 841.9 |
| 10 | Diga | 0 | 94.58 | 249.56 | 10.02 | 0.3 | 354.5 |
| 11 | Doreni | 0.24 | 170.58 | 264.01 | 25.27 | 0.6 | 460.7 |
| 12 | Gechi | 11.8 | 347.4 | 242.6 | 31.6 | 0.7 | 634.0 |
| 13 | Genji | 5.6 | 125.9 | 168.2 | 2.1 | 1.2 | 302.9 |
| 14 | Gimbi | 9.7 | 348.4 | 615.1 | 31.9 | 3.6 | 1008.8 |
| 15 | Haru | 4.56 | 200.58 | 267.16 | 10.48 | 0.5 | 483.3 |
| 16 | Homa | 1.1 | 47.4 | 46.7 | 0.4 | 0.0 | 95.6 |
| 17 | Hurumu | 0.5 | 106.4 | 230.1 | 125.0 | 4.0 | 465.9 |
| 18 | Lalo Asabi | 7.0 | 145.7 | 225.2 | 9.1 | 2.6 | 389.6 |
| 19 | Meko | 6.4 | 227.8 | 293.4 | 44.0 | 0.1 | 571.8 |
| 20 | Nole Kaba | 1.8 | 198.4 | 374.8 | 57.8 | 0.1 | 632.9 |
| 21 | Setema | 0.20 | 358.10 | 431.94 | 237.53 | 5.7 | 1033.5 |
| 22 | Seyo Nole | 32.7 | 295.7 | 282.1 | 1.7 | 2.5 | 614.6 |
| 23 | Yayu | 4.8 | 215.1 | 328.2 | 257.6 | 2.5 | 13265.7 |
| | Total | 177.1 | 5375.2 | 7246.0 | 1235.1 | 40.3 | 14073.8 |

## 4. Conclusion

Ethiopia's diverse topography and environmental conditions create a wide range of factors that influence rainfed wheat cultivation, including variable rainfall, temperature, soil properties and other key environmental variables. Wheat crop plays a vital role in the country's food security, economic stability, and employment, particularly benefiting smallholder farmers. Enhancing wheat production yields is closely linked to the suitability of cultivated land, which is influenced by a combination of biophysical, ecological, infrastructural, and environmental factors. Using advanced technologies such as Geographic Information Systems (GIS) and the Analytical Hierarchy Process (AHP), this study evaluated land suitability for wheat farming by processing and overlaying various environmental parameters. The analysis identified different land suitability classes within the southwestern parts of Ethiopia: approximately 177.1 km² (1.3%) was classified as highly suitable, 5375.2 km² (38.2%) as moderately suitable, 7246.0 km² (51.5%) as marginally suitable and 1235.1 km² (8.8%) as currently not suitable for rainfed wheat cultivation. As compare potential land suitability with selected districts, Sayo Nole district covered 32.7km² highly suitable for wheat cultivation. These findings are crucial for land resource management, providing actionable insights for optimizing wheat production. The results suggest that highly suitable areas should be prioritized for wheat cultivation to maximize yields, while unsuitable areas could be used for alternative crops. Further study could explore the impact of soil pH on wheat suitability and gather local community perspectives on land use in the study area.

## Supporting information

**S1 File. Inclusivity-in-global-research-questionnaire.**
(DOCX)

## Acknowledgments

The authors gratefully acknowledge the support and facilities Provided by Wollega University (Nekemte Campus of Natural and Computational Science and Shambu Campus Faculty of Resource Management and Economics), as well as the College of Economics and Management at Northwest A and F University, which were instrumental in conducting this study.

## Author contributions

**Conceptualization:** Bacha Gebissa Negeri.

**Formal analysis:** Bacha Gebissa Negeri, Mitiku Badasa Moisa.

**Methodology:** Mitiku Badasa Moisa.

**Project administration:** Mitiku Badasa Moisa.

**Software:** Mitiku Badasa Moisa.

**Supervision:** Xiuguang Bai.

**Validation:** Xiuguang Bai.

**Writing – original draft:** Bacha Gebissa Negeri.

**Writing – review & editing:** Xiuguang Bai.

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
