## [Decision Letter · Decision Letter 0]

Dear Dr. Bai,

Thank you for submitting your manuscript to PLOS ONE. After careful consideration, we feel that it has merit but does not fully meet PLOS ONE’s publication criteria as it currently stands. Therefore, we invite you to submit a revised version of the manuscript that addresses the points raised during the review process.

We look forward to receiving your revised manuscript.

Kind regards,

Nabin Rawal, PhD

Academic Editor

PLOS ONE

Journal Requirements:

3. Please include a complete copy of PLOS’ questionnaire on inclusivity in global research in your revised manuscript. Our policy for research in this area aims to improve transparency in the reporting of research performed outside of researchers’ own country or community. The policy applies to researchers who have travelled to a different country to conduct research, research with Indigenous populations or their lands, and research on cultural artefacts. The questionnaire can also be requested at the journal’s discretion for any other submissions, even if these conditions are not met.  Please find more information on the policy and a link to download a blank copy of the questionnaire here: https://journals.plos.org/plosone/s/best-practices-in-research-reporting. Please upload a completed version of your questionnaire as Supporting Information when you resubmit your manuscript.

“The authors declared no conflict of interests”

7. We note that Figures 1, 2, 3, 4, 5, 6 and 7 in your submission contain map/satellite images which may be copyrighted. All PLOS content is published under the Creative Commons Attribution License (CC BY 4.0), which means that the manuscript, images, and Supporting Information files will be freely available online, and any third party is permitted to access, download, copy, distribute, and use these materials in any way, even commercially, with proper attribution. For these reasons, we cannot publish previously copyrighted maps or satellite images created using proprietary data, such as Google software (Google Maps, Street View, and Earth). For more information, see our copyright guidelines: http://journals.plos.org/plosone/s/licenses-and-copyright.

a. You may seek permission from the original copyright holder of Figures 1, 2, 3, 4, 5, 6 and 7 to publish the content specifically under the CC BY 4.0 license.

Additional Editor Comments:

I would like to inform you that may be ready for further processing only after major revision. Please consider the enclosed reviewer's comments and make the necessary changes to your manuscript based on their advice.

To make it easier for the reviewers and editor to identify the revisions made during the second review round, it is essential to use ink in various colors to highlight the changes made to your manuscript. Your revised manuscript must be submitted with a covering letter that lists all the changes you have made to the work and addresses any feedback you have received from the editor and all reviewer(s). Please overview the journal guidelines for author for preparing the manuscript ready to the further processing. If you disagree with any comment or comments, kindly state your position and justification.

Thank you

Reviewers' comments:

Reviewer's Responses to Questions

**Comments to the Author**

1. Is the manuscript technically sound, and do the data support the conclusions?

Reviewer #1: Yes

Reviewer #2: Yes

2. Has the statistical analysis been performed appropriately and rigorously?

Reviewer #1: Yes

Reviewer #2: Yes

3. Have the authors made all data underlying the findings in their manuscript fully available?

Reviewer #1: Yes

Reviewer #2: No

4. Is the manuscript presented in an intelligible fashion and written in standard English?

Reviewer #1: Yes

Reviewer #2: Yes

Reviewer #1: This study evaluates land suitability for rain-fed wheat cultivation in southwestern Ethiopia using a GIS-based multi-criteria decision analysis, incorporating biophysical factors like soil, climate, and topography. The findings provide actionable insights for optimizing wheat production and food security, while recommending future studies to include additional ecological and socio-economic parameters for more comprehensive assessments. The following points should be addressed before further consideration and I will review the paper again after revisions:

Revise the manuscript for language clarity issues

Adjust the concept “rain fed” into “rainfed” in the title and the text

The title is extremely long. Substitute it with a shorter concise title. For instance, you can omit the scientific name of wheat. The word count in a good title should not exceed 20 words maximum.

In the abstract you state that “Wheat is one of the cereal crops rich in carbohydrates that are a source of energy for the human body. This energy is essential for body functions, physically activity, and maintaining overall health.” What does that have to do with your study? I see these sentences redundant and irrelevant to the subject under investigation. Revise the abstract to be more subject oriented, 2 lines for background, 2-3 lines for methodology and 6-8 lines for results, discussion, and conclusions.

Use better key words, e.g. Wheat, Rainfed agriculture, MCDA...etc.

Since the subject deals with rainfed agriculture, I expect to see a decent assessment of drought condition in Ethiopia during the past few decades. Drought is the main issue that drives researches similar to yours, highlighting areas suitable for rainfed agriculture and areas that are not. Still, I could not find a single mention to this issue. Make sure to add a section in the introduction highlighting drought issue and support it with references.

Highlight the importance of applying spatial rainfall anomaly indices (spatial rainfall data) and vegetation cover health data in highlighting agricultural drought (use this reference to support your statement: Javan, F. D., Samadzadegan, F., Toosi, A., & Tousikordkolaei, H. (2025). Spatial-temporal patterns of agricultural drought severity in the Lake Urmia Basin, Iran: A cloud-based integration of multi-temporal and multi-sensor remote sensing data. DYSONA-Applied Science, 6(2), 239-261. doi: 10.30493/das.2025.486806), and thereafter, selecting areas suitable for rainfed agriculture

At the end of the introduction you state that “This study was designed to solve the abovementioned gap by including identification of land suitability for wheat cultivation and provided future work for researchers.” This is not enough; a full description of research aims should be added.

Table 1: add links to the databases or use full name of the sources instead of abbreviations.

In the Descriptions of the study area, describe the climate of the area and wheat cultivation condition

The captions of all figures should be enhanced and more information should be incorporated. The readers should be able to understand the general purpose of the map (location, time, data…etc) without returning to the text.

The legend text should not cover any section of the map

Fig. 6: what does the gray selected districts map mean?

Reviewer #2: 1. What was the selection of criteria or factors based on? Please explain.

2. How was the soil investigated? From what depth was the sample taken? Was a profile dug? How many samples or profiles were investigated? Please indicate the location of the samples in the figure.

3. Why haven't other soil parameters such as lime, gypsum, salinity, alkalinity, etc. been investigated?

4. In the discussion section, it is better to make comparisons with the researches of others and the research done, and for the presented arguments, be sure to use valid and up-to-date references.

5. The advantages and disadvantages of the research done should be said.

6. Please provide appropriate and valid references for all provided relationships.

7. In the AHP method, did you use the opinions of relevant experts in the form of a questionnaire to determine the importance of the factors in the pairwise comparison matrix?

8. Please see the papers (https://doi.org/10.1007/s10661-022-10327-x,
https://doi.org/10.1080/03650340.2018.1549363,
https://doi.org/10.1016/j.geoderma.2017.09.012,
https://doi.org/10.1016/j.geoderma.2019.05.046,
https://doi.org/10.1080/00103624.2022.2072511;
https://doi.org/10.1080/03067319.2020.1746775;
https://doi.org/10.1007/s10661-022-10659-8;
https://doi.org/10.1080/00103624.2019.1626870) to improve the quality of the manuscript and use and add them to improve the quality of the manuscript, especially the introduction and discussion of the manuscript, description and interpretation of properties and select the criteria.

9. Please give the names of soils according to the WRB system.

10. What was the accuracy of the method used? By which criteria is the method evaluated?

11. Please check the grammar of the whole text with a native speaker and fix the errors.

**Do you want your identity to be public for this peer review?** For information about this choice, including consent withdrawal, please see our Privacy Policy

Reviewer #1: No

Reviewer #2: **Yes: ** Javad seyedmohammadi

---

## [Author Response · Author response to Decision Letter 1]

12 Apr 2025

Response Letter for each Reviewer

Dear Academic Editor

We sincerely appreciate the time and effort invested by the reviewers and the editor in evaluating our manuscript, Assessment of potential land suitability for rainfed wheat production using GIS and Multi criteria decision analysis in the Southwestern parts of Ethiopia. We are grateful for the constructive feedback and the opportunity to improve our work. In response to the major revision decision, we have carefully addressed all the comments provided. We have thoroughly revised the manuscript to ensure that the results are accurately reported, any overstated conclusions are rewritten, and the limitations of our study are clearly explained.

To facilitate the review process, we have prepared a point-by-point response outlining how each comment has been addressed. The revised manuscript has been uploaded alongside this response.

We appreciate your guidance and look forward to your further assessment.

Reviewer #1: This study evaluates land suitability for rain-fed wheat cultivation in southwestern Ethiopia using a GIS-based multi-criteria decision analysis, incorporating biophysical factors like soil, climate, and topography. The findings provide actionable insights for optimizing wheat production and food security, while recommending future studies to include additional ecological and socio-economic parameters for more comprehensive assessments. The following points should be addressed before further consideration and I will review the paper again after revisions:

Authors Response: Dear Reviewer, thank you very much for your valuable time and insightful comments on our manuscript, “Assessment of potential land suitability for rainfed wheat production using GIS and Multi criteria decision analysis in the Southwestern parts of Ethiopia." We truly appreciate your positive feedback and constructive suggestions, which have helped us, refine and improve our work. We acknowledge your concern about the need for a more careful and patient revision. We have now thoroughly reviewed the manuscript, ensuring that the content is more specific, concise, and well-structured. We have carefully addressed all your comments and revised the manuscript accordingly. We recommended for future studies to include additional ecological and socio-economic parameters. See highlighted text in the manuscript. We appreciate your thoughtful review and look forward to your further feedback on our revised version. Thank you again for your support and best wishes.

Revise the manuscript for language clarity issues

Authors Response: Dear Reviewer, Thank you very much for your valuable time and insightful comments. We improved the whole manuscript regarding to language and grammatical errors issues.

Adjust the concept “rain fed” into “rainfed” in the title and the text

Author’s response: Thank you very much for your valuable and constructive feedback. We have carefully addressed your comment and incorporated the necessary changes into the manuscript. The revised text is highlighted for your convenience.

The title is extremely long. Substitute it with a shorter concise title. For instance, you can omit the scientific name of wheat. The word count in a good title should not exceed 20 words maximum.

Author’s response: Thank you very much for your constructive comment. We improved the title. See highlighted text of the Title.

In the abstract you state that “Wheat is one of the cereal crops rich in carbohydrates that are a source of energy for the human body. This energy is essential for body functions, physically activity, and maintaining overall health.” What does that have to do with your study? I see these sentences redundant and irrelevant to the subject under investigation. Revise the abstract to be more subject oriented, 2 lines for background, 2-3 lines for methodology and 6-8 lines for results, discussion, and conclusions.

Author’s response: Thank you very much for your constructive comment. We improved the abstract. See highlighted text in the abstract.

Use better key words, e.g. Wheat, Rainfed agriculture, MCDA...etc.

Author’s response: Thank you very much for your constructive comment. We improved the keyword. See highlighted text of the keyword.

Since the subject deals with rainfed agriculture, I expect to see a decent assessment of drought condition in Ethiopia during the past few decades. Drought is the main issue that drives researches similar to yours, highlighting areas suitable for rainfed agriculture and areas that are not. Still, I could not find a single mention to this issue. Make sure to add a section in the introduction highlighting drought issue and support it with references.

Author’s response: Thank you very much for your constructive comment. We improved the introduction section as per your recommendation. See highlighted text in the manuscript.

Highlight the importance of applying spatial rainfall anomaly indices (spatial rainfall data) and vegetation cover health data in highlighting agricultural drought (use this reference to support your statement: Javan, F. D., Samadzadegan, F., Toosi, A., & Tousikordkolaei, H. (2025). Spatial-temporal patterns of agricultural drought severity in the Lake Urmia Basin, Iran: A cloud-based integration of multi-temporal and multi-sensor remote sensing data. DYSONA-Applied Science, 6(2), 239-261. doi: 10.30493/das.2025.486806), and thereafter, selecting areas suitable for rainfed agriculture.

Author’s response: Thank you very much for your constructive comment. We improved the introduction section as per your recommendation and we cited the recommended references. See highlighted text in the manuscript.

At the end of the introduction you state that “This study was designed to solve the abovementioned gap by including identification of land suitability for wheat cultivation and provided future work for researchers.” This is not enough; a full description of research aims should be added.

Author’s response: Thank you very much for your constructive comment. We improved the manuscript as per your recommendation. See highlighted text in the manuscript.

Table 1: add links to the databases or use full name of the sources instead of abbreviations.

Author’s response: Thank you very much for your constructive comment. We improved the Table 1as per your recommendation. See highlighted text in the Table 1.

In the Descriptions of the study area, describe the climate of the area and wheat cultivation condition

Author’s response: Thank you very much for your constructive comment. We improved the description of the study area as per your recommendation. See highlighted text in the manuscript.

The captions of all figures should be enhanced and more information should be incorporated. The readers should be able to understand the general purpose of the map (location, time, data…etc) without returning to the text. The legend text should not cover any section of the map

Author’s response: Thank you very much for your constructive comment. We improved the Caption of all Figures as per your recommendation. See highlighted text in the manuscript.

Fig. 6: what does the gray selected districts map mean?

Author’s response: Thank you very much for your constructive comment. We Removed it

Reviewer #2: 1. What was the selection of criteria or factors based on? Please explain.

Author’s response: Thank you very much for your constructive comment. We highlighted the selection of factors as per your recommendation. See highlighted text in the manuscript.

2. How was the soil investigated? From what depth was the sample taken? Was a profile dug? How many samples or profiles were investigated? Please indicate the location of the samples in the figure.

Author’s response: Thank you very much for your constructive comment. We improved it as per your recommendation. See highlighted text in the manuscript.

3. Why haven't other soil parameters such as lime, gypsum, salinity, alkalinity, etc. been investigated?

Author’s response: Thank you very much for your constructive comment. We improved it and recommended for future studies. See highlighted text in the manuscript.

4. In the discussion section, it is better to make comparisons with the researches of others and the research done, and for the presented arguments, be sure to use valid and up-to-date references.

Author’s response: Thank you very much for your constructive comment. We improved it as per your recommendation. See highlighted text in the manuscript.

5. The advantages and disadvantages of the research done should be said.

Author’s response: Thank you very much for your constructive comment. We improved it as per your recommendation. See highlighted text in the manuscript.

6. Please provide appropriate and valid references for all provided relationships.

Author’s response: Thank you very much for your constructive comment. We improved it as per your recommendation. See highlighted text in the manuscript.

7. In the AHP method, did you use the opinions of relevant experts in the form of a questionnaire to determine the importance of the factors in the pairwise comparison matrix?

Author’s response: Thank you very much for your constructive comment. We improved it as per your recommendation. See highlighted text in the manuscript.

8. Please see the papers (https://doi.org/10.1007/s10661-022-10327-x,
https://doi.org/10.1080/03650340.2018.1549363,
https://doi.org/10.1016/j.geoderma.2017.09.012,
https://doi.org/10.1016/j.geoderma.2019.05.046,
https://doi.org/10.1080/00103624.2022.2072511;
https://doi.org/10.1080/03067319.2020.1746775;
https://doi.org/10.1007/s10661-022-10659-8;
https://doi.org/10.1080/00103624.2019.1626870) to improve the quality of the manuscript and use and add them to improve the quality of the manuscript, especially the introduction and discussion of the manuscript, description and interpretation of properties and select the criteria.

Author’s response: Thank you sincerely for your insightful and constructive feedback. We have addressed your comment and ensured that all references have been properly cited as per your recommendation. The revised text is highlighted in the manuscript for your convenience.

9. Please give the names of soils according to the WRB system.

Author’s response: Thank you very much for your constructive comment. We improved it as per your recommendation. See highlighted text in the manuscript.

10. What was the accuracy of the method used? By which criteria is the method evaluated?

Author’s response: Thank you very much for your constructive comment. We improved it as per your recommendation. See highlighted text in the manuscript.

11. Please check the grammar of the whole text with a native speaker and fix the errors.

Author’s response: Thank you sincerely for your valuable and constructive feedback. We have carefully incorporated your recommendation into the manuscript. Please see the highlighted text for the revisions made.

---

## [Decision Letter · Decision Letter 1]

Assessment of potential land suitability for rainfed wheat production using GIS and Multi criteria decision analysis in the Southwestern parts of Ethiopia

PONE-D-25-03191R1

Dear Dr. Bai,

We’re pleased to inform you that your manuscript has been judged scientifically suitable for publication and will be formally accepted for publication once it meets all outstanding technical requirements.

Kind regards,

Nabin Rawal, PhD

Academic Editor

PLOS ONE

Additional Editor Comments (optional):

Thank you for addresing all the suggestion and comments. The manuscript can now be proceeded for further processing.

Reviewers' comments:

Reviewer's Responses to Questions

**Comments to the Author**

Reviewer #1: (No Response)

Reviewer #2: All comments have been addressed

2. Is the manuscript technically sound, and do the data support the conclusions?

Reviewer #1: (No Response)

Reviewer #2: Yes

3. Has the statistical analysis been performed appropriately and rigorously?

Reviewer #1: (No Response)

Reviewer #2: Yes

4. Have the authors made all data underlying the findings in their manuscript fully available?

Reviewer #1: (No Response)

Reviewer #2: Yes

5. Is the manuscript presented in an intelligible fashion and written in standard English?

Reviewer #1: (No Response)

Reviewer #2: Yes

Reviewer #1: The authors have made all necessary revisions. The manuscript is now ready for publication.

The authors have made all necessary revisions. The manuscript is now ready for publication.

Reviewer #2: The authors have made the necessary corrections. The text has been improved. Thanks for the corrections and improvements.

**Do you want your identity to be public for this peer review?** For information about this choice, including consent withdrawal, please see our Privacy Policy

Reviewer #1: No

Reviewer #2: No

---

## [Editor Report · Acceptance letter]

PONE-D-25-03191R1

PLOS ONE

Dear Dr. Bai,

I'm pleased to inform you that your manuscript has been deemed suitable for publication in PLOS ONE. Congratulations! Your manuscript is now being handed over to our production team.

Kind regards,

on behalf of

Dr. Nabin Rawal

Academic Editor

PLOS ONE